# Predictors of misconceptions, knowledge, attitudes, and practices of COVID-19 pandemic among a sample of Saudi population

**Mukhtiar Baig** [1][☺]*, **Tahir Jameel** [2][☺], **Sami H. Alzahrani** [3][☺], **Ahmad A. Mirza** [4][‡], **Zohair J. Gazzaz** [2][‡], **Tauseef Ahmad** [5][‡], **Fizzah Baig** [6][‡], **Saleh H. Almurashi** [7][‡]

1 Department of Clinical Biochemistry, Faculty of Medicine, Rabigh, King Abdulaziz University, Jeddah, Saudi Arabia, 2 Department of Internal Medicine, Faculty of Medicine, Rabigh, King Abdulaziz University, Jeddah, Saudi Arabia, 3 Department of Family Medicine, Faculty of Medicine, King Abdulaziz University, Jeddah, Saudi Arabia, 4 Department of Otolaryngology, Head and Neck Surgery, Faculty of Medicine in Rabigh, King Abdulaziz University, Jeddah, Saudi Arabia, 5 Department of Epidemiology and Health Statistics, School of Public Health, Southeast University, Nanjing, China, 6 Ziauddin Medical College, Ziauddin University, Karachi, Pakistan, 7 Faculty of Medicine, Rabigh, King Abdulaziz University, Jeddah, Saudi Arabia

☺ These authors contributed equally to this work.
‡ These authors also contributed equally to this work.
* drmukhtiarbaig@yahoo.com

**Data Availability Statement:** All relevant data are within the manuscript and its Supporting Information files.

## Abstract

This study intends to explore the predictors of misconceptions, knowledge, attitudes, and practices concerning the COVID-19 pandemic among a sample of the Saudi population and we also assessed their approaches toward its overall impact. This online cross-sectional survey was conducted at the Faculty of Medicine, Rabigh, King Abdulaziz University (KAU) in Jeddah, Saudi Arabia (SA). Participants were approached via social media (SM), and 2006 participants (953 [47.5%] females and 1053 [52.5%] males) were included in this study. SM was the leading source of information for 43.9% of the study participants. Most of the participants had various misconceptions such as "females are more vulnerable to develop this infection, rinsing the nose with saline and sipping water every 15 minutes protects against Coronavirus, flu and pneumonia vaccines protect against this virus." About one-third of participants (31.7%) had self-reported disturbed social, mental, and psychological wellbeing due to the pandemic. Many participants became more religious during this pandemic. Two-thirds of the study participants (68.1%) had good knowledge scores. Attitudes were highly positive in 93.1%, and practice scores were adequate in 97.7% of the participants. Participants' educational status was a predictor of high knowledge scores. Male gender and divorced status were predictors of low practice scores, and aged 51–61 years, private-sector jobs, and student status were predictors of high practice scores. Being Saudi was a predictor of a positive attitude, while the male gender and divorced status were predictors of a negative attitude. Higher education was a predictor of good concepts, while the older age and businessmen were predictors of misconceptions. Overall, our study participants had good knowledge, positive attitudes, and good practices, but several myths were

**Funding:** The author(s) received no specific funding for this work.

**Competing interests:** The authors have declared that no competing interests exist.

also prevalent. Being a PhD and a Saudi national predicted high knowledge scores and positive attitudes, respectively. A higher education level was a predictor of good concepts, and students, private-sector jobs, and aged 51–61 years were predictors of high practice scores. Study participants had good understanding of the effects of this pandemic.

## Introduction

In recent years, coronaviruses have become a major health hazard worldwide, and they have caused considerable human morbidity. In a short period of time, the novel Coronavirus Disease 2019 (COVID-19) has spread globally. This infection has been transmitted to 213 countries and territories worldwide and infected 25,925,003 people causing 860,857 deaths (as of September 3, 2020) [1]. It has caused deterioration of everything from worldwide economies to people's social lives. Initially, the COVID-19 pandemic was viewed with ignorance, mayhem, repudiation, and fright. However, it spread at an unbelievably rapid pace, infecting thousands of people worldwide. Most countries had to lock down their cities, and at that point, people took serious notice and started taking precautionary measures [2, 3].

Saudi Arabia has witnessed a variety of disturbing Coronavirus epidemics over the last several years, such as Severe Respiratory Distress Syndrome- Coronavirus (SARS-CoV) in 2002 and Middle Eastern Respiratory Distress Syndrome-Coronavirus (MERS-CoV) in 2012 [4, 5]. SARS-CoV-2 (the virus that causes COVID-19) is a newer member of this family and it has become a pandemic within a very short time. This highly infectious virus has severely affected SA, and cases have been recorded in almost all regions [6]. Authorities took drastic preventive and curative measures, including a phase-by-phase lockdown and curfew imposition during the evening in almost all major cities [7]. When these measures proved to be less effective, the curfew was extended to nearly 24 hours with a brief break for buying essential commodities. Until now (September 3, 2020), the number of positive COVID-19 cases has increased to 317,486, with 3,956 deaths due to this disease [1].

The control of communicable diseases depends mainly on the local population's knowledge, attitudes, practices, and behavior [8]. The strict observance of precautionary measures to avoid spreading this disease to the masses is key to controlling it. People have been overburdened by the influx of information from different resources, especially from SM; thus, people are confused and anxious to find accurate knowledge [9, 10]. It is imperative to understand public awareness, attitudes, commitment, and compliance with and acceptance of measures that affect their daily lives in a number of ways, especially mentally, socially, and physically. This understanding could be achieved by analyzing the general public knowledge, attitudes, and practices [11, 12]. In this context, the present survey explored the predictors of misconceptions, knowledge, attitudes, and practices concerning the COVID-19 pandemic among a sample of the Saudi population. We also assessed their approaches toward the overall effects of this pandemic. Our results could help update awareness campaigns accordingly and provide baseline data for devising future pandemics policies.

## Methods

The present cross-sectional questionnaire-based survey was conducted at the Faculty of Medicine, Rabigh, KAU, Jeddah, SA, after obtaining ethical approval from the Unit of Biomedical Ethics of the University (Ref No. 187–20). The Raosoft sample size calculator calculated the sample size; considering the margin of error at 5%, confidence level of 95%, and population size 3000000, the required sample size was 385. However, we submitted the questionnaire to

3000 individuals. The sample size was expanded owing to the predicted lower turnover in an online questionnaire. The convenience sample technique was used, and no monetary benefit was offered to any participants. An online questionnaire was constructed with the help of the World Health Organization (WHO) myth-buster document and a published study [13, 14]. This questionnaire was converted to a Google document, and participants were approached using SM (Facebook, WhatsApp, Twitter, and others). A brief description of the research and a request for participation were presented at the beginning of the questionnaire. Completion of the online questionnaire was considered to indicate consent for participation in the survey.

The questionnaire was translated and back-translated (English/Arabic) by two bilingual experts, and the questionnaire was modified according to their suggestions. Two senior faculty members validated the questionnaire. In order to assess the convenience and interpretation of the questionnaire, we carried out a pilot study on 35 participants from the general population and modified the questionnaire accordingly. The reliability of the questionnaire was 0.81 (Cronbach's alpha). People younger than 18 years of age and residing outside Jeddah were excluded from the study. We included only residents of Jeddah to keep the study focused on a cosmopolitan area.

Our questionnaire had several parts, and the first part consisted of demographic questions like age, education, job, marital status, etc. In SA, general education comprises kindergarten, six years of primary school, and three years each of intermediate and high school. Higher education in SA is four years in the humanities and social sciences and five or six years in the medical, engineering, and pharmacy fields. Fourteen knowledge questions, four attitudes, six practices, 19 misconceptions, and six impact questions were also on the questionnaire. Questions regarding knowledge, attitudes, misconceptions, and impact of the outbreak were true/false/not sure types, while the practice questions were yes/no/sometimes types. One score was awarded for true, and zero for false and not sure, and an individual score less than 50% (1–7 score), 51%–75% (8–10 score), and 76%–100% (11–14 score) were considered poor, moderate, and good, respectively. For attitudes, marking ranged from −4 to +4 (true answer +1 and false and not sure −1). An individual's positive score indicated a positive attitude, while a negative or zero scores indicated a negative attitude. The practice score ranged from 0 to 12 (yes = 2 points, sometimes = 1, and no = 0), and a score of ≥6 was considered adequate while <6 was considered inadequate. The misconception questions score ranged from 1 to 19 (correct [true] answer = 1 score, wrong [false] answer = 0 score, not sure = 0 score) and scores ≤50% (1–9) were considered to be poor concepts, and individuals with a >50% score (10–19) were considered high scorers, indicated good concepts.

## Statistical analysis

The collected data were analyzed using Statistical Package for Social Sciences version 26 (SPSS-26). It includes an integrated set of computer programs that allow users to read questionnaire survey data and other sources to modify data in different ways to generate a wide variety of statistical analyses or reports.

For various variables, the descriptive analysis is represented as frequency and percentage. We used chi-square test to investigate the comparison between demographic variables. The dependent variables (knowledge and practice) were considered numeric variables, and independent variables (age, education, nationality, job, marital status) were categorical variables, so we used multiple linear regression analysis to compute association. Dependent variables (misconception and attitude) were considered binominal categorical, and independent variables (age, education, nationality, job, marital status) were also categorical; thus, logistic regression analysis was applied to explore association. Additionally, $p < 0.05$ was considered to be significant.

# Results

A total of 2117 participants completed the questionnaire, and after removing incomplete responses, 2006 participants ([47.5%] females and [52.5%] males) were included in the study. The general characteristics of the study participants are shown in Table 1. Among study participants, the sources for seeking COVID-19 information are shown in Fig 1.

The participants' responses regarding knowledge, attitudes, and practices are shown in S1 Table.

A few common misconceptions were "females are more vulnerable to develop this infection" (56.2%), "sipping water every 15 minutes protects against Coronavirus" (43.5%), and "flu and pneumonia vaccines protect against this virus" (50.9%). About half of the respondents (46.3%) were terrified of COVID-19. About one-third (31.7%) of the study participants had self-reported disturbed social, mental, and psychological wellbeing resulting from the pandemic's circumstances. Many participants became more religious (S2 Table).

Two-thirds of the study participants (68%) had good knowledge of COVID-19, and 26.6% had moderate knowledge. The attitude of the majority of the participants (93.1%) was highly positive, and the practice score was adequate in 97.7% of the participants. Two-thirds of the study participants (66%) had misconceptions (score <50 [poor]), while one-third of the participants (34%) had good concepts (score >50 [good]) as shown in Fig 2.

Significant differences were found in knowledge scores according to age groups (p = 0.037) and educational status (p < 0.001). A significant difference in attitudes was observed according to gender (females were more positive), nationality (Saudis were more positive), education (except Master's degrees were more positive), and marital status (married were more positive). A significant difference in the practice score was observed between age groups (p = 0.041) and

**Table 1. General characteristics of study participants.**

| Variables | | N | % |
|---|---|---|---|
| Gender | Male | 1046 | 52.1 |
| | Female | 960 | 47.9 |
| Age (years) | 18–28 | 771 | 38.4 |
| | 29–39 | 524 | 26.1 |
| | 40–50 | 499 | 24.9 |
| | 51–61 | 175 | 8.7 |
| | >61 | 37 | 1.8 |
| Nationality | Saudi | 1710 | 85.2 |
| | Non-Saudi | 296 | 14.8 |
| Education level | Primary school | 22 | 1.1 |
| | High school graduate | 360 | 17.9 |
| | College | 1375 | 68.5 |
| | Master's | 176 | 8.8 |
| | Ph.D. | 66 | 3.3 |
| Job | Government job | 808 | 40.3 |
| | Private-sector job | 296 | 14.8 |
| | Business owner | 63 | 3.1 |
| | Housewife | 293 | 14.6 |
| | Student | 538 | 26.8 |
| Marital Status | Married | 1172 | 58.4 |
| | Unmarried | 771 | 38.4 |
| | Divorced | 55 | 2.7 |

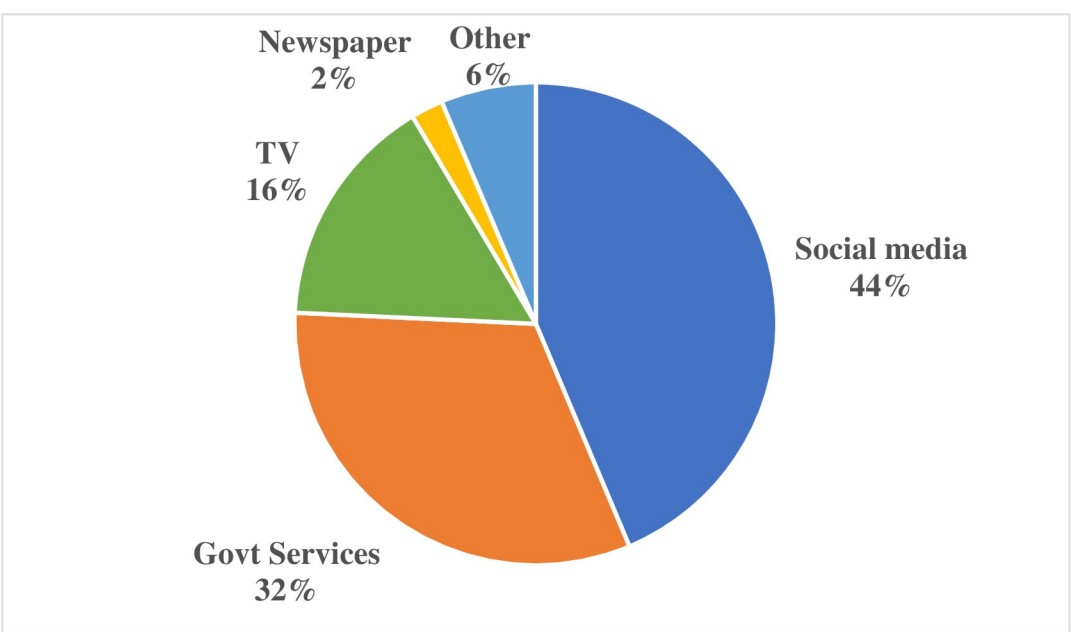

**Fig 1. Information sources for study participants.**

gender (p <0.001). More participants in the age groups 18–28 years and 29–39 years showed a good understanding of pandemic concepts compared to the other age groups (p <0.001), and males also showed good concepts as compared to females (p <0.001). Highly educated people (college, Master's degrees, PhD) showed good concepts compared to people with primary school and high school education (p <0.001). Students and people in private-sector and government jobs had good concepts compared to housewives and businessmen (p <0.001). According to marital status, unmarried people had better concepts than married and divorced people (p <0.001), as shown in Table 2.

Multiple linear regression analysis showed that the participants' educational status (Ph.D.) was a predictor of high knowledge scores. Male gender and divorced status were predictors of low practice scores, and aged 51–61 years, private-sector jobholders, and students were predictors of the high practice scores (Table 3).

Binary regression analysis revealed that being a Saudi national was a predictor of having positive attitudes, while the male gender and divorced status were predictors of negative attitudes. Higher education was a predictor of good concepts, while older age and business owners were predictors of misconceptions (Table 4).

## Discussion

Awareness and a positive response by society are critical to the successful handling of emergencies such as the COVID-19 pandemic.

### Source of information

Among our participants, the leading source of information about COVID-19 was SM, followed by government websites, television, newspapers, and others. Our results are similar to another study [15]. Meier et al. also reported that television, newspapers, official health websites, and SM were the most frequently used information sources [16]. SM has extensive "health misinformation," often described as information that contradicts existing evidence from medical specialists

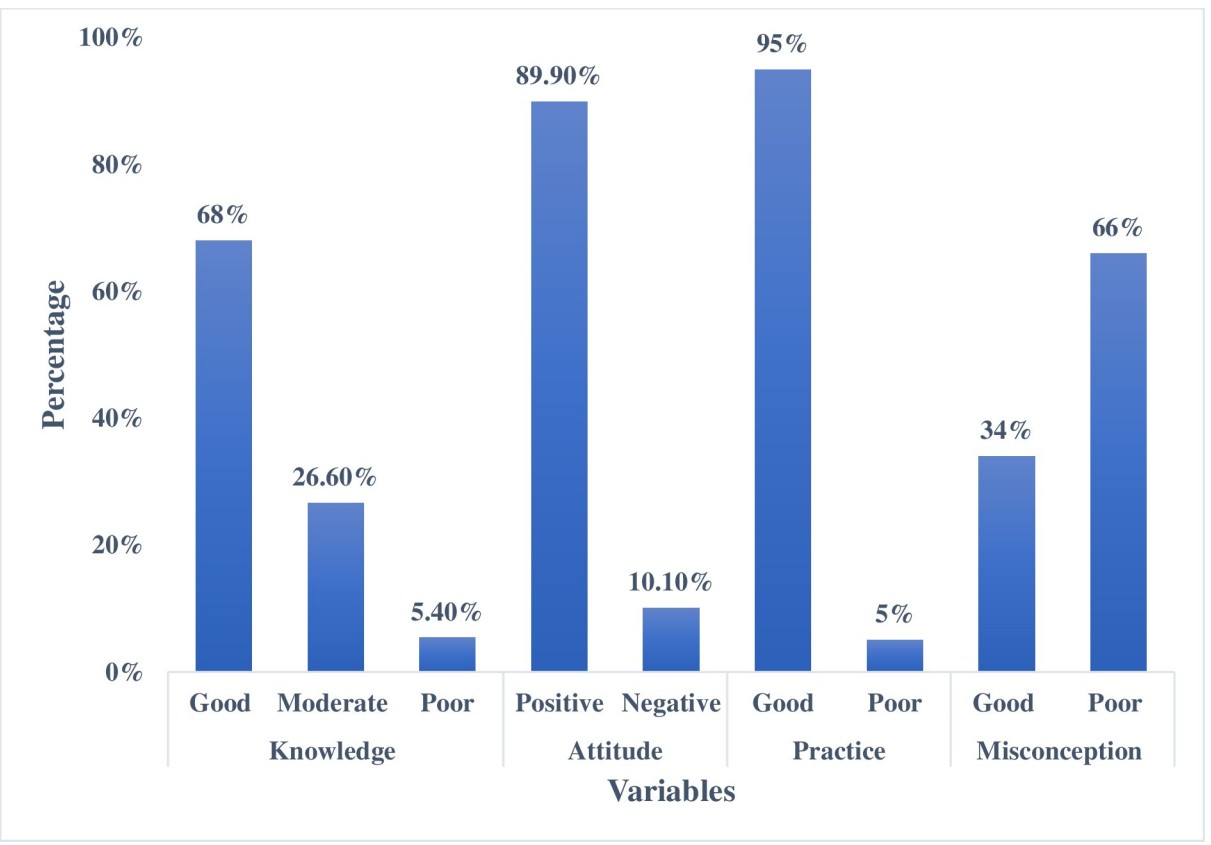

**Fig 2. Knowledge, attitudes, practices, and misconceptions scores of study participants.** Knowledge score <50% = poor knowledge, 50–75% score = moderate knowledge, >75% score = good knowledge. A positive score indicates a positive attitude, while negative and zero scores indicate negative attitudes. A practice score of ≥6 was considered adequate, and <6 was considered inadequate. A misconception score ≤50% = poor concepts, while > 50% = good concepts.

[17]. Therefore, information seekers must search for medical information from reliable resources, such as the WHO, the Centers for Disease Control (CDC), and their ministry of health (MOH) portals. It is worrying that most of our study participants were seeking information from SM during this pandemic. However, one-third of the study population were using government services for finding information. The MOH in SA has been working very efficiently since the beginning of the pandemic, and its portal regularly updates information regarding COVID-19.

## Misconceptions

Several misconceptions were present among two-thirds of the study participants. Limited data is available about the association of misconceptions with demographic variables. Our results revealed that high education levels (Master's degrees and PhD) were predictors of good concepts while being a businessman and old age were predictors for having misconceptions. Similar to our results, a few other studies have also reported misconceptions among their study participants [18, 19]. An Australian and a multinational study reported that uncertainties and misconceptions about COVID-19 were widespread among the general public [15, 18]. Such misconceptions and misinformation could be an obstacle against taking appropriate precautionary measures and positive behavior changes among the masses. Identification of accurate knowledge about a disease is essential for risk-reduction behavior. Most interventions have concentrated on information propagation as an imperative phase in reducing disease risk [20].

**Table 2. Comparison of knowledge, attitude, practice, and misconceptions scores according to socio-demographic variables.**

| Variables | | Knowledge | | | Attitude | | Practice | | Misconception | |
|---|---|---|---|---|---|---|---|---|---|---|
| | | Poor | Moderate | Good | Negative | Positive | Inadequate | Adequate | ≤ 50 score | > 50 score |
| | | N (%) | N (%) | N (%) | N (%) | N (%) | N (%) | N (%) | N (%) | N (%) |
| Age (years) | 18–28 | 37 (5) | 162 (22) | 536 (72.9) | 55 (7.1) | 716 (92.9) | 22 (2.9) | 743 (97.1) | 403 (55.4) | 324 (44.6) |
| | 29–39 | 29 (5.7) | 142 (28) | 337 (66.3) | 44 (8.4) | 480 (91.6) | 17 (3.3) | 499 (96.7) | 316 (64.6) | 173 (35.4) |
| | 40–50 | 23 (5) | 138 (29.8) | 302 (65.2) | 33 (6.6) | 466 (93.4) | 7 (1.4) | 486 (98.64) | 370 (77.4) | 108 (22.6) |
| | 51–61 | 10 (6.3) | 52 (32.5) | 98 (61.3) | 6 (3.4) | 169 (32.5) | 0 (0) | 174 (100) | 132 (78.1) | 37 (21.9) |
| | >61 | 1 (3.2) | 11 (35.5) | 19 (61.3) | 1 (2.7) | 36 (97.3) | 0 (0) | 37 (100) | 29 (82.9) | 6 (17.1) |
| p-value | 0.037* | | | | 0.183 | | 0.041 | | < 0.001* | |
| Gender | Male | 60 (6) | 269 (27) | 668 (67.0) | 89 (8.5) | 957 (91.5) | 41 (3.9) | 1004 (96.1) | 614 (61.6) | 382 (38.4) |
| | Female | 40 (4.4) | 236 (26.2) | 624 (69.3) | 50 (5.3) | 901 (94.7) | 5 (0.5) | 928 (99.5) | 630 (70.8) | 363 (29.5) |
| p-value | 0.259 | | | | 0.004* | | < 0.001* | | < 0.001* | |
| Nationality | Saudi | 79 (4.9) | 429 (26.4) | 1114 (68.7) | 102 (6) | 1608 (94) | 43 (2.5) | 1649 (97.5) | 545 (33.8) | 1614 (100) |
| | Non-Saudi | 21 (7.6) | 76 (27.6) | 178 (64.7) | 36 (12.5) | 251 (87.5) | 3 (1.1) | 281 (98.9) | 101 (36.7) | 275 (100) |
| p-value | 0.130 | | | | <0.001* | | 0.125 | | 0.339 | |
| Education | Primary school | 2 (11.1) | 7 (38.9) | 9 (50) | 1 (4) | 21 (95.5) | 0 (0) | 22 (100) | 18 (94.7) | 1 (5.3) |
| | College | 58 (4.4) | 318 (24.3) | 932 (71.3) | 86 (6.3) | 1289 (93.7) | 27 (2) | 1334 (98) | 811 (62.3) | 491 (37.7) |
| | High school graduate | 19 (5.6) | 132 (38.9) | 188 (55.5) | 16 (4.4) | 344 (95.6) | 8 (2.2) | 348 (97.8) | 281 (81.4) | 64 (18.6) |
| | Master | 15 (8.9) | 33 (19.6) | 120 (71.4) | 30 (17) | 146 (83) | 8 (4.6) | 166 (95.4) | 102 (61.8) | 63 (38.2) |
| | Ph.D | 6 (9.4) | 15 (23.4) | 43 (67.2) | 6 (9.1) | 60 (90.9%) | 3 (4.6) | 62 (95.4) | 36 (59) | 25 (41) |
| p-value | < 0.001 | | | | < 0.001* | | 0.153 | | < 0.001* | |
| Job | Government job | 40 (5.2) | 215 (28) | 513 (66) | 52 (6.4) | 756 (93.6) | 23 (2.9) | 779 (97.1) | 520 (68) | 245 (32) |
| | Private sector job | 16 (5.6) | 76 (26.4) | 196 (68.1) | 23 (7.8) | 273 (92.2) | 4 (1.4) | 291 (98.6) | 173 (62.0) | 106 (38) |
| | Business owner | 3 (5.8) | 18 (34.6) | 31 (59.6) | 6 (9.5) | 57 (90.5) | 3 (4.7) | 61 (95.3) | 52 (82.5) | 11 (17.5) |
| | Housewife | 16 (5.9) | 84 (31.2) | 169 (62.8) | 16 (5.5) | 277 (94.5) | 1 (0.4) | 283 (99.6) | 220 (80) | 55 (20) |
| | Student | 25 (4.8) | 112 (21.5) | 383 (73.7) | 41 (7.6) | 497 (92.4) | 15 (2.8) | 518 (97.2) | 278 (54.7) | 230 (45.3) |
| p-value | 0.090 | | | | 0.614 | | 0.055 | | < 0.001* | |
| Marital Status | Married | 60 (5.4) | 308 (27.8) | 739 (66.8) | 74 (6.3) | 1098 (93.7) | 19 (1.6) | 1141 (98.4) | 797 (71.9) | 312 (28.1) |
| | Unmarried | 36 (4.9) | 179 (24.1) | 527 (71.0) | 56 (7.3) | 715 (92.7) | 25 (3.3) | 740 (96.7) | 406 (55.7) | 323 (44.3) |
| | Divorced | 4 (8.3) | 18 (37.5) | 26 (54.2) | 9 (16.4) | 46 (83.6) | 2 (3.8) | 50 (96.1) | 40 (76.9) | 12 (23.1) |
| p-value | 0.084 | | | | 0.015* | | 0.052 | | < 0.001* | |

Total number of responses is not same in all categories because of few missing responses.

Educated people have more exposure and access to knowledge, which explains why they had clear concepts. We could not find any plausible explanation for the association of misconceptions with old age and businessmen. Continuous efforts are needed to clarify the peoples' misconceptions, and in some cases, this process should involve religious leaders, especially in Muslim countries. The most important parameter is sustained awareness campaigns by government and non-governmental organizations (NGOs) to dispel these myths because some of these myths are harmful to people.

## Impact

About half of the respondents were terrified of COVID-19, and about one-third of the study participants had self-reported disturbed social, mental, and psychological wellbeing. Most participants said that they realized the importance of life because of this pandemic, and one-third were committed to becoming more religious. Most of our respondents stated that this

**Table 3. Multiple linear regression model for predictors of knowledge and practice.**

| Variables | Knowledge [a] | | | Practice [a] | | |
|---|---|---|---|---|---|---|
| | B | P-value | 95% CI for B | | B | P-value | 95% CI for B |
| | | | Lower Bound -Upper Bound | B | P-value | Lower Bound -Upper Bound |
| Age (years) | | | | | | |
| 29–39 | -0.087 | 0.605 | -3.012–1.685 | 0.056 | 0.707 | -0.235–0.347 |
| 40–50 | -0.041 | 0.825 | -2.942–2.202 | 0.169 | 0.295 | -0.148–0.486 |
| 51–61 | -0.271 | 0.232 | -5.243–1.085 | 0.462 | **0.020** | 0.074–0.851 |
| >61 | 0.091 | 0.813 | -4.849–5.971 | 0.609 | 0.061 | -0.028–1.247 |
| Gender | | | | | | |
| Male | -0.197 | 0.051 | -2.826 - -0.002 | -1.115 | **0.000** | -1.290 - -0.940 |
| Nationality | | | | | | |
| Saudi | 0.118 | 0.402 | -0.158–0.395 | -0.217 | 0.083 | -0.463–0.028 |
| Educational status | | | | | | |
| College | 0.852 | 0.067 | -0.375–12.637 | -0.405 | 0.234 | -1.071–0.262 |
| High school graduate | 0.343 | 0.468 | -4.065–9.154 | -0.340 | 0.328 | -1.021–0.342 |
| Master | 0.738 | 0.129 | -1.626–11.980 | -0.612 | 0.092 | -1.322–0.099 |
| Ph.D. | 1.215 | **0.021** | 1.055–15.707 | -0.240 | 0.554 | -1.036–0.556 |
| Job status | | | | | | |
| Private job | 0.028 | 0.852 | -2.082–1.876 | 0.413 | **0.002** | 0.152–0.675 |
| own business | -0.356 | 0.210 | -6.717–1.174 | 0.275 | 0.242 | -0.186–0.735 |
| housewife | -0.227 | 0.156 | -4.004–0.391 | 0.031 | 0.826 | -0.247–0.310 |
| Student | 0.180 | 0.305 | -1.261–3.632 | 0.337 | **0.028** | 0.036–0.639 |
| Marital status | | | | | | |
| Unmarried | -0.043 | 0.780 | -2.501–1.865 | -0.179 | 0.188 | -0.446–0.087 |
| Divorced | -0.559 | 0.052 | -7.996–0.083 | -0.503 | **0.045** | -0.994 - -0.011 |

[a] Numeric variable

pandemic's worst impact would be on the country's economic conditions, followed by the healthcare system and the peoples' financial status.

Another study reported inclinations of people to gravitate toward religion during this pandemic issue. They also embraced a healthy lifestyle, stayed away from the mass gatherings, and prayed at home instead of attending mosques [21]. Gros et al. also stated raised awareness and concerns among their study participants regarding the economic situation during this pandemic [22]. A Chinese study reported a similar impact of this pandemic among its population [23]. Holmes et al. emphasized taking steps to tackle mental health problems, such as anxiety and depression, during the COVID-19 pandemic [24].

## Knowledge

In our study, two-thirds of the participants had good knowledge, and one-quarter had moderate knowledge scores. More participants in the younger age and educated groups had high knowledge scores. The participants' educational status (PhD) was a predictor of high knowledge scores. Our results are similar to several other studies [18, 19, 25]. An Australian study described the general public's good knowledge in Australia, but they found knowledge gaps for a few questions like "some people have natural immunity to virus, letters from china can spread the virus, the virus was genetically engineered, the virus was human-made" [15]. In contrast to our results, a Pakistani study reported a low knowledge score among the general public [26].

**Table 4. Multiple logistic model for predictors of attitude and misconception.**

| Variables | Attitude | | | Misconception | | |
|---|---|---|---|---|---|---|
| | OR | P-value | 95% CI for B<br>Lower limit—Upper limit | OR | P-value | 95% CI for B<br>Lower limit—Upper limit |
| 18–28 (years) | Reference | | | | | |
| 29–39 (years) | 0.777 | 0.447 | 0.405–1.491 | 0.857 | 0.406 | 0.597–1.232 |
| 40–50 (years) | 1.061 | 0.873 | 0.511–2.206 | 0.483 | **0.001** | 0.319–0.730 |
| 51–61 (years) | 2.522 | 0.089 | 0.868–7.328 | 0.432 | **0.002** | 0.254–0.737 |
| >61 (years) | 3.571 | 0.244 | 0.420–30.36 | 0.330 | **0.039** | 0.115–0.947 |
| Female | Reference | | | | | |
| Male | 0.503 | **0.002** | 0.328–0.770 | 1.236 | 0.063 | 0.988–1.546 |
| Non-Saudi | Reference | | | | | |
| Saudi | 2.354 | **0.001** | 1.451–3.818 | 0.791 | 0.145 | 0.577–1.084 |
| Primary school | Reference | | | | | |
| College | 0.843 | 0.870 | 0.109–6.515 | 7.512 | 0.053 | 0.974–57.924 |
| High school graduate | 1.139 | 0.903 | 0.140–9.245 | 3.103 | 0.281 | 0.396–24.312 |
| Master | .254 | 0.195 | 0.032–2.022 | 9.657 | **0.031** | 1.224–76.185 |
| Ph.D | 0.536 | 0.582 | 0.058–4.946 | 12.092 | **0.021** | 1.467–99.661 |
| Govt job | Reference | | | | | |
| Private job | 1.193 | 0.562 | 0.657–2.167 | 0.988 | 0.944 | 0.706–1.382 |
| Own business | 0.647 | 0.374 | 0.247–1.693 | 0.295 | **0.004** | 0.127–0.684 |
| Housewife | 0.674 | 0.276 | 0.331–1.371 | 0.671 | 0.052 | 0.449–1.003 |
| Student | 0.578 | 0.119 | 0.290–1.152 | 1.074 | 0.711 | 0.736–1.566 |
| Married | Reference | | | | | |
| Unmarried | 0.965 | 0.908 | 0.532–1.753 | 1.244 | 0.202 | 0.889–1.741 |
| Divorced | 0.287 | **0.002** | 0.129–0.639 | 1.181 | 0.613 | 0.620–2.248 |

Our results found no significant association between participants' knowledge scores and other variables. In contrast to our results, a few studies found a significant association between knowledge and demographic characteristics, such as age, gender, and occupation [13, 19]. The awareness level was much higher among educated people, a common finding in other studies [27, 28]. In a bi-national survey, most of the respondents had a good knowledge score regarding COVID-19, and knowledge scores were associated with the 18–39 years age group, college/bachelor's education, and the participants' background [29]. In the Ethiopian population, knowledge scores and practicing behaviors were not up to the mark regarding COVID-19 [34]. While a Pakistani study reported a significant association between good knowledge scores and adequate attitudes and practices [26].

Similar to our study, most Malaysian populations had good knowledge, positive attitudes, and good practicing behavior regarding COVID-19. The researchers credit this awareness to an effective campaign by the health authorities and government [11]. A survey from Arabic-speaking Middle Eastern countries identified several gaps in public knowledge about COVID-19 and proposed health education to amend their knowledge [18]. Similar suggestions were provided by a Chinese study [13].

## Attitudes

Most of our study participants had positive attitudes and believed that society has a social responsibility to implement safety measures to control the spread of this infection. Malaysian and Chinese studies have also reported positive attitudes among participants for similar

questions [13, 25]. Our study participants' positive attitudes may be attributed to the MOH's excellent campaign for the Saudi population's awareness. They send daily awareness messages on mobile phones in different languages and have launched a mobile application to identify COVID-19 symptoms. Newspapers and television are also disseminating information regarding preventive measures. Such good attitudes among people were also attributed to the government's efforts to mitigate viral transmission in Chinese and Malaysian studies [13, 25].

Being a Saudi was a predictor of having positive attitudes, while the male gender and divorced status were predictors of negative attitudes. One reason for this difference could be that Saudi nationals have a better living style and more exposure to awareness campaigns in the local language on SM and local TV networks. Most of the expatriates belong to the working class, and they are not highly educated. Thus, they had relatively insufficient knowledge regarding the disease compared to Saudi nationals, which is also reflected in their attitudes. Interestingly, this finding has also been highlighted in other studies in which females showed more concern and positivity toward their families and society with respect to any infectious pandemic [30, 31].

Education and marriage modify individual responses resulting in responsible attitudes and overall positiveness [32]. This finding was observed in our study with some exceptions. One interesting finding was that divorced status association with a careless and rather negative attitude toward COVID-19. Nasser et al. mentioned similar results in their participants [18].

## Practices

Our study found good practices, and these results are similar to a few other studies [19, 21]. Being male and having a divorced status were predictors of low practice scores, and aged 51–61 years, private-sector jobs, and students were predictors of the high practice scores. Our findings are similar to recently carried out investigations that described females' more responsible role than males of the same age group [13, 18]. Our results are more or less similar to those of a Malaysian study that also found good practices among the general Malaysian population toward COVID-19 [25]. A Japanese study reported good practices among study participants, particularly in females and older participants [33].

Several explanations for the study participants' good practices can be described, including implementing strict curfew and lockdown across the country. People were not allowed to go to their neighboring areas and other cities during lockdown breaks. Because of the rapid spread and thousands of deaths worldwide, so much apprehension has already been generated among the populace. All mass media sources have been full of COVID-19 news, and SM has also been swiftly disseminating information. Thus, the knowledge and practice scores were good, and attitudes were more positive among our study participants.

## Suggestions and recommendations

Our results suggest that most of the study population have responded to this pandemic situation in a very responsible way; however, certain sections of society need more education and mass awareness programs. Because the development of a vaccine against COVID-19 will take time, people will have to learn to live with COVID-19 in society. Recently, WHO officials have announced that it is the probability that "this coronavirus may become just another endemic virus in our communities, and this virus may never go away" [34]. No country can afford a ban on all commercial activities and closure of air routes and its borders for extended periods. Therefore, we should continue working while observing the WHO instructions for strict precautionary measures, such as personal hygiene, good sneezing and coughing etiquette, and frequent washing of our hands with soap to protect ourselves and others. A clear policy should be

constituted to deal with the people's psychological and mental wellbeing in this pandemic. Our study results can be used by policymakers cautiously in information campaigns on COVID-19 by the MOH/public health authorities and the mass media.

## Limitations

There are a few limitations to our study, including the use of an online questionnaire, so we could not reach the section of society that didn't use the internet. We used a convenience sampling technique, and in such studies, the respondents' biases cannot be ignored. Besides, our study sample was not representative of the total population and all segments of society. Moreover, the study design employed was cross-sectional. Hence the results need to be reviewed with caution.

## Conclusion

Overall, our study participants had good knowledge, positive attitudes, and good practices; however, several myths were also prevalent. Having a PhD and being a Saudi national predicted high knowledge scores and positive attitudes, respectively. Higher education was a predictor of good concepts, and students, private-sector jobs, and aged 51–61 years were predictors of high practice scores. Study participants had good understanding of the effects of this pandemic. It seems that despite all the measures, the only chance of success against this highly infectious disease is coordinated and consistent efforts to increase public concern against the disease. Moreover, people should follow the government-issued standard operating procedures when performing their daily tasks.

## Supporting information

**S1 Table. Study participants' knowledge, attitudes, and practices regarding COVID-19 pandemic.**
(DOCX)

**S2 Table. Study participants' misconceptions and impact of COVID-19 pandemic.**
(DOCX)

**S1 Data.**
(SAV)

## Acknowledgments

We are thankful to our medical students, Mahmood Abdullah Eid, Jinan Hikmat Msallati, Abdulelah Mugbil Hajer Alnafie, Nouf Khaleel Althagafi, Sultan Abdu Madkhali, Abdulrahman Omar Alzahrani, for their help in collecting data.

## Author Contributions

**Conceptualization:** Mukhtiar Baig, Sami H. Alzahrani, Tauseef Ahmad.

**Data curation:** Tahir Jameel, Fizzah Baig.

**Formal analysis:** Mukhtiar Baig.

**Methodology:** Sami H. Alzahrani, Ahmad A. Mirza, Tauseef Ahmad, Saleh H. Almurashi.

**Project administration:** Sami H. Alzahrani, Ahmad A. Mirza, Zohair J. Gazzaz, Saleh H. Almurashi.

**Resources:** Tahir Jameel, Ahmad A. Mirza, Saleh H. Almurashi.

**Supervision:** Zohair J. Gazzaz.

**Writing – original draft:** Tahir Jameel.

**Writing – review & editing:** Mukhtiar Baig, Zohair J. Gazzaz, Tauseef Ahmad, Fizzah Baig.

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
