## [Decision Letter · Decision Letter 0]

9 Oct 2020

PONE-D-20-26398

Predictors of misconceptions, knowledge, attitudes, and practices of COVID-19 pandemic among a sample of Saudi population

PLOS ONE

Dear Dr. Baig,

Thank you for submitting your manuscript to PLOS ONE. After careful consideration, we feel that it has merit but does not fully meet PLOS ONE’s publication criteria as it currently stands. Therefore, we invite you to submit a revised version of the manuscript that addresses the points raised during the review process.

We look forward to receiving your revised manuscript.

Kind regards,

Ritesh G. Menezes, M.B.B.S., M.D., Diplomate N.B.

Academic Editor

PLOS ONE

Journal Requirements:

Reviewers' comments:

Reviewer's Responses to Questions

**Comments to the Author**

1. Is the manuscript technically sound, and do the data support the conclusions?

Reviewer #1: Yes

Reviewer #2: Yes

Reviewer #3: Partly

2. Has the statistical analysis been performed appropriately and rigorously? 

Reviewer #1: Yes

Reviewer #2: Yes

Reviewer #3: No

3. Have the authors made all data underlying the findings in their manuscript fully available?

Reviewer #1: Yes

Reviewer #2: Yes

Reviewer #3: Yes

4. Is the manuscript presented in an intelligible fashion and written in standard English?

Reviewer #1: Yes

Reviewer #2: Yes

Reviewer #3: No

5. Review Comments to the Author

Reviewer #1: The article is well written and is comprehensive overall. However, there are some minor corrections needed to be addressed.

The first line in the abstract should be 'The study intends to explore'

In the introduction, line 71, mentioning words like 'severely' 'deadly' makes the sentence look very dramatic. either remove severely or deadly. Again in line 82 and 94 words like 'war against COVID-19' and 'battle against COVID-19' is very dramatic. Use of more subtle words is recommended.

Line 240, in the misconceptions part of discussion, it should be 'Limited data is available..', not are.

Again in line 364, I think the use of the word 'deadly' is not appropriate as COVID-19 has a spectrum of presentations, from being asymptomatic to having pneumonia etc. it does not always cause critical illness. Line 368, use COVID-19 patients instead of corona-infected patients.

Reviewer #2: Comments to the authors

This manuscript focuses on a relevant issue in the current scenario and the research question is novel. Authors have written the objectives of the study clearly. Abstract gives the summary of the manuscript in a concise manner.

However, there are few concerns with respect to this manuscript which need to be addressed by the authors.

Abstract

• Line 52 – Authors can mention some of the prevailing myths in the abstract as well.

Methodology What were the inclusion/exclusion criteria?

• Line 125 – what score was given to the responses as not sure?

• Did the scores have weightage based on the type of questions?

• Line 146 – why did authors recruit 2117 participants while the estimated sample size was 385? In that case, there is no need to mention about sample size estimation, I believe.

• It would be appropriate to mention the socio-economic status of the participants instead of mentioning education and occupation separately (table 1).

• The category Misconception could be classified as Present or Absent instead of Good and Poor.

Results

• Presentation of results within the text is not appropriate. For example …A few common misconceptions were “females are more vulnerable to develop this infection 1067(56.2%), instead it is better to mention only the percentages within the brackets as

A few common misconceptions were “females are more vulnerable to develop this infection (56.2%)

• The significant predictors could be mentioned in bold within the tables so that it will easy for the readers to understand the associations between the variables.

• How did the authors assess the mental and psychological wellbeing of the participants? This aspect needs to be mentioned clearly in the methodology.

Discussion

• Line 270-281 mentions only about other study findings. Authors should highlight their findings and compare it with other study findings and discuss the similarities or differences observed.

• Line 281 – Is it participants’ reasons or reasons for knowledge among participants. Meaning of both would be entirely different. Grammatical errors need to be corrected in many parts of the manuscript.

• Line 300-301 –What is the relevance of this sentence in the given context? It is not clear what the authors are trying to convey.

Reviewer #3: • The authors have discussed the study findings which are relevant in the current context.

• The authors need to acknowledge that the sample was not representative & the study design employed was cross-sectional. Hence the results need to be reviewed with caution.

• Overall English language editing needs to be done & grammatical errors need to be corrected.

• Predictors & Impact are not appropriate terms for cross –sectional study design. Need to be corrected.

• Study participants had a proper realization of the impact of this pandemic. – needs to be corrected

• people causing 860,857 mortalities – mortalities is not correct usage – needs to be rectified.

• Cases have been recorded in almost all the regions with pocketing of cases in multiple cities – needs to be reworded.

• on pg 10, 82-95 lines in the later part of introduction is repetitive, needs to be condensed & revised.

• Complete the online questionnaire was considered consent for participation in the survey.-reword. Was there a check box available for the participants to check - I consent

• pilot study consisting of 35 people from the general public was conducted to determine the convenience and comprehension of the questionnaire, - -reword

• what was the basis of grading the scores as good, inadequate poor? Was it based on the consensus of experts or the authors or based on published literature?

• The frequency and percentages were computed for different variables. A chi-squared test was used to explore the comparison between different variables. – needs to be more explicit

• Table 1 – class intervals for age group to be kept uniform

• Define primary school, college etc in educational level categories as footnotes or in methods section as educational classification varies between countries.

• one-third 637 (31.7%) of the study participants had disturbed social, mental, and psychological wellbeing – how was this assessed?

• It is preferable not to repeat results in tables & text.

• Table 2,3,4 titles to be reworded. The table titles need to be stand alone & self explanatory. Preferable to depict total n in the table title.

• Table 3,4 B values not required, 95% CI can be depicted as -0.235, 0.347 in a single column

• Table 4 – preferable to write it as unadjusted OR

• The statistically significant values can be highlighted in bold in tables.

• Univariate analysis should be followed by multivariate analysis. The tables have described it in reverse order.

• The same dependant variables used in univariate analysis have to be used for multivariate analysis & different variables cannot be used as shown in Table 3 & 4.

• Initial part of the discussion is mainly focussing on use of SM for source of knowledge on COVID – needs to be condensed. Discussion should compare & contrast the current study findings with available literature

• one-third of the study participants had disturbed social, mental, and psychological wellbeing as mentioned in discussion has not been discussed in the results section

• categorization of misconceptions, basis for the classification of misconceptions is important to discuss in results rather than only p value or 95% CI

• mental health/illness was not assessed in the present study nor was it part of the objectives – hence it is not wise to put it in discussion & state it in conclusions

• Our results are similar to several other studies [8,22,23]. An Australian study described the general public’s good knowledge in Australia, but they had knowledge gaps for a few questions [17] – discussion needs to be more explicit.

• knowledge scores and practicing behaviours were not up to the mark in terms of fighting the spread of COVID-19 [34]. –reword the statement

• Our study’s high-level positive attitude may be attributed to the excellent campaign conducted by the MOH for the Saudi population’s awareness – high level positive attitude – words may be used in moderation

• Practices: Almost all the participants attained adequate scores regarding adopting protective measures. Our study participants maintained social distancing and avoided meeting with friends and relatives. Most of the participants frequently used soap for handwashing and used face masks outside the home – most of these statements are pertaining to results of the present study – these need to be described in the results section & not in the discussion.

• Overall the results & discussion have to be reworked on. The discussion section needs to be condensed. Most of the results stated in the discussion section needs to be moved to results section.

• 12,22,26,27,37 citation of references is incomplete & not in standard format

• Fig 2 – axis & axis title to be provided

• Our study results can be used by policymakers to set priorities in information campaigns on COVID-19 by the MOH/public health authorities and the mass media. – is not appropriate as the study is not based on a representative sample & as authors have acknowledged that only internet users were included. In addition, COVID situation being very dynamic, the study recommendations have to be in moderation.

6. PLOS authors have the option to publish the peer review history of their article (what does this mean?). If published, this will include your full peer review and any attached files.

Reviewer #1: **Yes: **Tehlil Rizwan

Reviewer #2: No

Reviewer #3: No

---

## [Author Response · Author response to Decision Letter 0]

28 Oct 2020

Review Comments to the Author

Reviewer #1: 

Q. The article is well written and is comprehensive overall. However, there are some minor corrections needed to be addressed.

The first line in the abstract should be 'The study intends to explore'

Reply:

We have modified the sentence as suggested.

Q. In the introduction, line 71, mentioning words like 'severely' 'deadly' makes the sentence look very dramatic. either remove severely or deadly. 

Reply:

We have modified the sentences as suggested.

Q. Again in line 82 and 94 words like 'war against COVID-19' and 'battle against COVID-19' is very dramatic. Use of more subtle words is recommended.

Reply:

We modified the sentences as suggested.

Q. Line 240, in the misconceptions part of discussion, it should be 'Limited data is available..', not are.

Reply:

We have modified the sentence as suggested.

Q. Again in line 364, I think the use of the word 'deadly' is not appropriate as COVID-19 has a spectrum of presentations, from being asymptomatic to having pneumonia etc. it does not always cause critical illness. 

Reply:

We have modified the sentence as suggested.

Q. Line 368, use COVID-19 patients instead of corona-infected patients.

Reply:

We have modified the sentence as suggested.

Reviewer #2: Comments to the authors

This manuscript focuses on a relevant issue in the current scenario and the research question is novel. Authors have written the objectives of the study clearly. Abstract gives the summary of the manuscript in a concise manner. However, there are few concerns with respect to this manuscript which need to be addressed by the authors.

Q. Abstract

• Line 52 – Authors can mention some of the prevailing myths in the abstract as well.

Reply:

We have incorporated the few myths in line number 40-43, as suggested.

Q. Methodology What were the inclusion/exclusion criteria?

Reply:

Inclusion and exclusion criteria are given in line 124-126 in methodology.

Q. • Line 125 – what score was given to the responses as not sure?

Reply:

One score was awarded for true, and zero for false and not sure.

Q• Did the scores have weightage based on the type of questions?

Reply:

No. All questions had equal weightage.

Q• Line 146 – why did authors recruit 2117 participants while the estimated sample size was 385? In that case, there is no need to mention about sample size estimation, I believe.

Reply:

We have removed the sample size calculation, as suggested.

Q• It would be appropriate to mention the socio-economic status of the participants instead of mentioning education and occupation separately (table 1).

Reply

It's a nice suggestion, but we have used these variables in regression analysis separately, so it is difficult to replace them with socioeconomic status. So we sorry that we are unable to incorporate this suggestion.

Q• The category Misconception could be classified as Present or Absent instead of Good and Poor.

Reply:

Actually, if we will write misconceptions absent or present, then it will give a different impression.

There were several misconceptions present and after calculating their scores, we categorized them as good concepts and poor concepts.

Q. Results

• Presentation of results within the text is not appropriate. For example …A few common misconceptions were “females are more vulnerable to develop this infection 1067(56.2%), instead it is better to mention only the percentages within the brackets as

A few common misconceptions were “females are more vulnerable to develop this infection (56.2%)

Reply:

Thank you for the suggestion. We have incorporated this suggestion and changed the results accordingly. 

Q. • The significant predictors could be mentioned in bold within the tables so that it will easy for the readers to understand the associations between the variables.

Reply:

Thank you for the suggestion. We have mentioned the significant predictors values in bold in the table. 

Q • How did the authors assess the mental and psychological wellbeing of the participants? This aspect needs to be mentioned clearly in the methodology.

Reply:

We didn’t use any inventory for this. This was the self-reported disturbed social, mental, and psychological wellbeing due to the pandemic. We have mentioned this in the methods.

Discussion

• Line 270-281 mentions only about other study findings. Authors should highlight their findings and compare it with other study findings and discuss the similarities or differences observed.

Reply:

We have modified the discussion as suggested and removed a few sentences, and added a new reference.

Q• Line 281 – Is it participants' reasons or reasons for Knowledge among participants. Meaning of both would be entirely different.

Reply: 

The sentence has been modified.

Q. Grammatical errors need to be corrected in many parts of the manuscript.

Reply:

A professional editing service has done the language editing.

Q• Line 300-301 –What is the relevance of this sentence in the given context? It is not clear what the authors are trying to convey.

Reply:

The sentence has been removed.

Reviewer #3: • 

The authors have discussed the study findings which are relevant in the current context.

• The authors need to acknowledge that the sample was not representative & the study design employed was cross-sectional. Hence the results need to be reviewed with caution.

Reply:

Thank you for the suggestion. We have incorporated this suggestion in the limitations. 

• Overall English language editing needs to be done & grammatical errors need to be corrected.

Reply:

A professional editing service has done the language editing.

Q• Predictors & Impact are not appropriate terms for cross –sectional study design. Need to be corrected.

Reply:

Multiple Linear Regression and Logistic Regression analyses are considered predictive analysis. Therefore, we have used the term predictive. In our study impact, related questions were self-reflective questions that's why we had used these terminologies. In literature, a few other studies found that they have used word predictive in cross-sectional studies. 

Q• Study participants had a proper realization of the impact of this pandemic. – needs to be corrected

Reply:

We have modified the sentence as suggested.

Q• people causing 860,857 mortalities – mortalities is not correct usage – needs to be rectified.

Reply:

We have replaced the mortalities with deaths.

Q• Cases have been recorded in almost all the regions with pocketing of cases in multiple cities – needs to be reworded.

Reply:

We have modified the sentence as suggested.

Q• on pg 10, 82-95 lines in the later part of introduction is repetitive, needs to be condensed & revised.

Reply:

We have removed a few sentences.

Q• Complete the online questionnaire was considered consent for participation in the survey.-reword. Was there a check box available for the participants to check - I consent

Reply:

We are sorry that there was no check box for consent. 

Q• pilot study consisting of 35 people from the general public was conducted to determine the convenience and comprehension of the questionnaire, - -reword

Reply:

We have modified the sentence as suggested.

Q• what was the basis of grading the scores as good, inadequate poor? Was it based on the consensus of experts or the authors or based on published literature?

Reply:

It was based on the consensus of the researchers.

Q• The frequency and percentages were computed for different variables. A chi-squared test was used to explore the comparison between different variables. – needs to be more explicit

Reply:

We have modified the sentence as suggested.

Q• Table 1 – class intervals for age group to be kept uniform

Reply:

Thank you for the suggestion. We have modified the age group intervals and these are uniform now.

Q• Define primary school, college etc in educational level categories as footnotes or in the methods section as educational classification varies between countries. 

Reply:

We have written a description of SA education system in the methods.

Q• one-third 637 (31.7%) of the study participants had disturbed social, mental, and psychological wellbeing – how was this assessed?

Reply:

We didn’t use any inventory for this. This is the respondents self-reported statement. We have mentioned this in the methods.

Q• It is preferable not to repeat results in tables & text.

Reply: 

We have reduced a few results from the text. 

Q• Table 2,3,4 titles to be reworded. The table titles need to be stand alone & self explanatory. Preferable to depict total n in the table title.

Reply:

The title has been modified as per suggestion.

Q• Table 3,4 B values not required, 95% CI can be depicted as -0.235, 0.347 in a single column

Reply:

In Table 3, Knowledge and practice are numeric variables and B values indicate the change in knowledge and practice score by one-unit change in dependents variables used in multiple regression model. In Table 4, B values have been removed.

Q• Table 4 – preferable to write it as unadjusted OR

Reply:

In table 4, adjusted OR was given because we used multiple logistic regression model.

Q• The statistically significant values can be highlighted in bold in tables.

Reply:

We have modified as suggested.

Q• Univariate analysis should be followed by multivariate analysis. The tables have described it in 

reverse order.

Reply:

Univariate analysis data was not given in tables.

Q• The same dependant variables used in univariate analysis have to be used for multivariate analysis & different variables cannot be used as shown in Table 3 & 4.

Reply:

Table 3 & 4 have no link with each other. In table 3, dependent variables are Knowledge and practice, and both were taken as numeric, while in table 4, dependent variables are attitude and misconception and both were taken as binary variables (categorical variables). 

Q• Initial part of the discussion is mainly focussing on use of SM for source of Knowledge on COVID – needs to be condensed. Discussion should compare & contrast the current study findings with available literature 

Reply:

We have modified as suggested.

Q• one-third of the study participants had disturbed social, mental, and psychological wellbeing as mentioned in discussion has not been discussed in the results section

Reply:

This has been described in the result section in line number 208-210.

Q• categorization of misconceptions, basis for the classification of misconceptions is important to discuss in results rather than only p value or 95% CI.

Reply:

In the results section, in the legends of figure 2 it has been described.

Q• mental health/illness was not assessed in the present study nor was it part of the objectives – hence it is not wise to put it in discussion & state it in conclusions

Reply:

We have removed most of the statements and references related to mental health. 

Q• Our results are similar to several other studies [8,22,23]. An Australian study described the general public's good Knowledge in Australia, but they had knowledge gaps for a few questions [17] – discussion needs to be more explicit.

Reply:

We have explained this point.

Q• knowledge scores and practicing behaviours were not up to the mark in terms of fighting the spread of COVID-19 [34]. –reword the statement

Reply:

We have modified as suggested.

Q• Our study’s high-level positive attitude may be attributed to the excellent campaign conducted by the MOH for the Saudi population’s awareness – high level positive attitude – words may be used in moderation

Reply:

We have modified the sentence as suggested.

Q• Practices: Almost all the participants attained adequate scores regarding adopting protective measures. Our study participants maintained social distancing and avoided meeting with friends and relatives. Most of the participants frequently used soap for handwashing and used face masks outside the home – most of these statements are pertaining to results of the present study – these need to be described in the results section & not in the discussion.

Reply:

We have deleted these sentences from the discussion as suggested.

Q• Overall the results & discussion have to be reworked on. The discussion section needs to be condensed. Most of the results stated in the discussion section needs to be moved to results section.

Reply:

We have modified as suggested. The discussion section has been condensed, and most of the description of the results have been removed from the discussion.

Q• 12,22,26,27,37 citation of references is incomplete & not in standard format

Reply:

Few references have been updated and two are preprints, so we have given their ODI.

Q• Fig 2 – axis & axis title to be provided

Reply:

We have modified the figure as suggested.

Q• Our study results can be used by policymakers to set priorities in information campaigns on COVID-19 by the MOH/public health authorities and the mass media. – is not appropriate as the study is not based on a representative sample & as authors have acknowledged that only internet users were included. Besides, COVID situation being very dynamic, the study recommendations have to be in moderation.

Reply:

We have modified our recommendations, as suggested.

---

## [Decision Letter · Decision Letter 1]

19 Nov 2020

PONE-D-20-26398R1

Predictors of misconceptions, knowledge, attitudes, and practices of COVID-19 pandemic among a sample of Saudi population

PLOS ONE

Dear Dr. Baig,

Thank you for submitting your manuscript to PLOS ONE. After careful consideration, we feel that it has merit but does not fully meet PLOS ONE’s publication criteria as it currently stands. Therefore, we invite you to submit a revised version of the manuscript that addresses the points raised during the review process.

Please submit your revised manuscript by 25-November-2020. Please include the following items when submitting your revised manuscript:

We look forward to receiving your revised manuscript.

Kind regards,

Ritesh G. Menezes, M.B.B.S., M.D., Diplomate N.B.

Academic Editor

PLOS ONE

Reviewers' comments:

Reviewer's Responses to Questions

**Comments to the Author**

1. If the authors have adequately addressed your comments raised in a previous round of review and you feel that this manuscript is now acceptable for publication, you may indicate that here to bypass the “Comments to the Author” section, enter your conflict of interest statement in the “Confidential to Editor” section, and submit your "Accept" recommendation.

Reviewer #1: All comments have been addressed

Reviewer #2: All comments have been addressed

Reviewer #3: All comments have been addressed

2. Is the manuscript technically sound, and do the data support the conclusions?

Reviewer #1: Yes

Reviewer #2: Yes

Reviewer #3: Yes

3. Has the statistical analysis been performed appropriately and rigorously? 

Reviewer #1: Yes

Reviewer #2: Yes

Reviewer #3: Yes

4. Have the authors made all data underlying the findings in their manuscript fully available?

Reviewer #1: Yes

Reviewer #2: (No Response)

Reviewer #3: Yes

5. Is the manuscript presented in an intelligible fashion and written in standard English?

Reviewer #1: Yes

Reviewer #2: No

Reviewer #3: Yes

6. Review Comments to the Author

Reviewer #1: As suggested previously, the manuscript was very comprehensive except a few minor changes which have been addressed by the authors in this current revision.

Reviewer #2: Authors have addressed all the queries raised by the reviewers and edited the manuscript accordingly.

Reviewer #3: The authors have addressed the queries previously raised. Few minor corrections & grammatical errors to be corrected.

In the section on Statistical analysis

The collected data were analyzed using SPSS-26. – provide the details of the software.

To investigate the comparison between demographic variables, a chi-squared test was used. It should be chi-square test. The test was employed to assess the association between variables – to reword the statement.

Sample size calculation, basis & rationale for the numbers included needs to be described.

Table 2. Comparison of knowledge, attitude, practice, and misconceptions scores according to age, gender, nationality, education, job and marital status (n=2006).

n mentioned in results & n in title is different.

reword the title & mention it as socio-demographic variables, instead of naming all the variables in the title.

n (%) to be used in column header in table 2, so that % symbol need not be used in every cell of the table.

Table 3 & 4 - zero needs to precede the decimal point in the tables for better readability.

Pg 17 lines-60-62

People have been scared and have mental distress, and these factors are understandable because health is a serious matter for most people. The sentence may be deleted

Pg 19 – lines 113 -114

Practices - Our study found good practicing behaviour – to be reworded as good practices & not good practicing behaviour

Pg 20 - lines 150-152

Conclusion

Overall, our study participants had good knowledge, highly positive attitudes, and excellent

practicing behavior; however, several myths were also prevalent. – to moderate the sentences & avoid usage of highly positive, excellent practice etc

7. PLOS authors have the option to publish the peer review history of their article (what does this mean?). If published, this will include your full peer review and any attached files.

Reviewer #1: No

Reviewer #2: No

Reviewer #3: No

---

## [Author Response · Author response to Decision Letter 1]

21 Nov 2020

Review Comments to the Author

Reviewer #1: As suggested previously, the manuscript was very comprehensive except a few minor changes which have been addressed by the authors in this current revision.

Reply:

Thanks

Reviewer #2: Authors have addressed all the queries raised by the reviewers and edited the manuscript accordingly.

Reply:

Thanks

Reviewer #3: The authors have addressed the queries previously raised. Few minor corrections & grammatical errors to be corrected.

Reply:

Several grammatical and punctuation errors have been corrected. 

In the section on Statistical analysis. The collected data were analyzed using SPSS-26. – provide the details of the software.

Reply:

Details have been added.

To investigate the comparison between demographic variables, a chi-squared test was used. It should be chi-square test. The test was employed to assess the association between variables – to reword the statement.

Reply:

Corrections have been done as suggested.

Sample size calculation, basis & rationale for the numbers included needs to be described.

Reply:

Sample size calculation and rationale have been added as suggested.

Table 2. Comparison of knowledge, attitude, practice, and misconceptions scores according to age, gender, nationality, education, job and marital status (n=2006).

n mentioned in results & n in title is different.

Yes, you are right. That's why we didn't include the total number of participants in the table's title in the original manuscript. After the suggestion of one of the reviewers, we included the number of participants. We have deleted the number of study subjects from the title of the study to remove the discrepancy. To further clarify, we have included the sentence "Total number of responses is not same in all categories because of few missing responses” below the table. This discrepancy is due to the fact that few people didn't reply to a few questions.

Reword the title & mention it as socio-demographic variables, instead of naming all the variables in the title.

Reply:

Corrections have been done as suggested and socio-demographic variables have replaced all variable names.

n (%) to be used in column header in table 2, so that % symbol need not be used in every cell of the table.

Reply:

Corrections have been done as suggested.

Table 3 & 4 - zero needs to precede the decimal point in the tables for better readability.

Reply:

Corrections have been done as suggested.

Pg 17 lines-60-62

People have been scared and have mental distress, and these factors are understandable because health is a serious matter for most people. The sentence may be deleted

Reply:

The sentence has been deleted as suggested.

Pg 19 – lines 113 -114

Practices - Our study found good practicing behaviour – to be reworded as good practices & not good practicing behaviour

Reply:

The sentence has been reworded as suggested.

Pg 20 - lines 150-152

Conclusion

Overall, our study participants had good knowledge, highly positive attitudes, and excellent practicing behavior; however, several myths were also prevalent. – to moderate the sentences & avoid usage of highly positive, excellent practice etc

Reply:

The sentence has been reworded as suggested.

---

## [Editor Report · Decision Letter 2]

24 Nov 2020

Predictors of misconceptions, knowledge, attitudes, and practices of COVID-19 pandemic among a sample of Saudi population

PONE-D-20-26398R2

Dear Dr. Baig,

We’re pleased to inform you that your manuscript has been judged scientifically suitable for publication and will be formally accepted for publication once it meets all outstanding technical requirements.

Kind regards,

Ritesh G. Menezes, M.B.B.S., M.D., Diplomate N.B.

Academic Editor

PLOS ONE
---

## [Editor Report · Acceptance letter]

1 Dec 2020

PONE-D-20-26398R2 

Predictors of misconceptions, knowledge, attitudes, and practices of COVID-19 pandemic among a sample of Saudi population 

Dear Dr. Baig:

I'm pleased to inform you that your manuscript has been deemed suitable for publication in PLOS ONE. Congratulations! Your manuscript is now with our production department. 

Kind regards, 

on behalf of

Prof. Dr. Ritesh G. Menezes 

Academic Editor

PLOS ONE